# Predicting risk of suicidal behaviour after initiation of selective serotonin reuptake inhibitors in children, adolescents and young adults: protocol for development and validation of clinical prediction models

Tyra Lagerberg  ,[1,2] Suvi Virtanen,[1] Ralf Kuja-Halkola,[1] Clara Hellner,[3] Paul Lichtenstein,[1] Seena Fazel,[2,4] Zheng Chang[1]

¹Department of Medical Epidemiology and Biostatistics, Karolinska Institutet, Stockholm, Sweden
²Department of Psychiatry, Warneford Hospital, University of Oxford, Oxford, UK
³Department of Clinical Neuroscience, Centre for Psychiatry Research, Karolinska Institutet, Stockholm, Sweden
⁴Oxford Health NHS Foundation Trust, Oxford, UK

**Correspondence to**
Dr Tyra Lagerberg;
tyra.lagerberg@ki.se

## ABSTRACT

**Introduction** There is concern regarding suicidal behaviour risk during selective serotonin reuptake inhibitor (SSRI) treatment among the young. A clinically useful model for predicting suicidal behaviour risk should have high predictive performance in terms of discrimination and calibration; transparency and ease of implementation are desirable.

**Methods and analysis** Using Swedish national registers, we will identify individuals initiating an SSRI aged 8–24 years 2007–2020. We will develop: (A) a model based on a broad set of predictors, and (B) a model based on a restricted set of predictors. For the broad predictor model, we will consider an ensemble of four base models: XGBoost (XG), neural net (NN), elastic net logistic regression (EN) and support vector machine (SVM). The predictors with the greatest contribution to predictive performance in the base models will be determined. For the restricted predictor model, clinical input will be used to select predictors based on the top predictors in the broad model, and inputted in each of the XG, NN, EN and SVM models. If any show superiority in predictive performance as defined by the area under the receiver-operator curve, this model will be selected as the final model; otherwise, the EN model will be selected. The training and testing samples will consist of data from 2007 to 2017 and from 2018 to 2020, respectively. We will additionally assess the final model performance in individuals receiving a depression diagnosis within 90 days before SSRI initiation.

The aims are to (A) develop a model predicting suicidal behaviour risk after SSRI initiation among children and youths, using machine learning methods, and (B) develop a model with a restricted set of predictors, favouring transparency and scalability.

**Ethics and dissemination** The research is approved by the Swedish Ethical Review Authority (2020–06540). We will disseminate findings by publishing in peer-reviewed open-access journals, and presenting at international conferences.

## STRENGTHS AND LIMITATIONS OF THIS STUDY

⇒ The study will make use of a large and nationally representative data linkage of Swedish registers, covering virtually any Swedish resident who initiated selective serotonin reuptake inhibitor treatment during the study period.

⇒ We will compare different machine learning models and approaches to selecting predictors, in order to explore the performance of an ensemble prediction model using machine learning methods in Swedish register data, and to develop a clinically applicable model with a restricted set of predictors where transparency and scalability are favoured.

⇒ We will follow the Transparent Reporting of a multivariable prediction model for Individual Prognosis or Diagnosis (TRIPOD) guidelines in the reporting of our study results.

⇒ The prediction models will be developed and validated in Swedish register data, and the extent to which they can be generalised is unclear: further work to validate the model in other national contexts will be necessary.

## INTRODUCTION

Suicide is among the leading causes of death among younger individuals.[1] Meanwhile, suicide is often accompanied, and/or driven, by underlying psychiatric disorders—mood disorders in particular.[1] Compared with the general population, depressed individuals have been found to have around a 20 times higher rate of suicide.[2] Antidepressants are the main pharmacological treatment option for mood disorders—taken by over 12% of the population in the USA[3]—and selective serotonin reuptake inhibitors (SSRIs) are the most common antidepressant type in many countries.[3 4] They are also the first-line

pharmacological treatment for depression among children and adolescents in many countries, including in Sweden.[5]

Although commonly prescribed, there are concerns about the risk of suicide during SSRI treatment among children and adolescents, based on evidence from randomised control trials[6] and observational studies.[7] Questions on whether and in what circumstances this association indicates causation remain.[7] Regardless of causality, producing a model to assess which individuals, out of those initiating an SSRI, are more or less likely to engage in suicidal behaviour could be an important aid in clinical decision-making in terms of determining which individuals will require more clinical oversight. Models that predict suicidal behaviour risk within short time frames are expected to be most clinically relevant.[8] However, applying a risk prediction model represents only one aspect of a more comprehensive approach necessary to prevent suicidal behaviour.[9]

Many studies have developed prediction models for suicide or suicidal behaviour in different populations.[10] However, no prior study has predicted suicidal behaviour among children, adolescents and young adults who initiate an SSRI.

Some previous prediction studies have incorporated a wide variety of available predictors in order to harness the potential of large routinely collected electronic health records.[11–13] For example, a previous study using Swedish register data was able to predict suicide following a psychiatric inpatient visit with a relatively high level of discrimination performance.[12] However, this approach results in large models that may not be practical to implement in real-life settings where clinicians do not have access to routinely collected data. In such a setting, a clinician might want a restricted number of predictors to input into a risk prediction model that is simple, scalable and transparently operationalised – for example as an online tool.[14 15] Previous research also suggests that clinicians want to understand why specific individuals are assigned a specific risk score, in particular when the risk classification is counterintuitive,[16] which may favour the latter type of model.

Given the high baseline risk among many under consideration for SSRI-treatment, and the possibility of an elevated risk observed early during treatment,[8] we therefore aim to predict the risk of suicidal behaviour within 30 days of SSRI initiation in a population of children, adolescents and young adults, aged 8–24 years. We do this in two ways: by applying a broad predictor set and relatively complex ensemble machine-learning approach, as compared with a clinically informed approach using a restricted set of predictors and favouring a more transparent model. It should be noted that our aim is not to determine which individuals should or should not be treated with an SSRI, but to identify whom, among those who do initiate an SSRI, is at particularly elevated risk of suicidal behaviour. Given that children under age 18 are routinely treated for psychiatric disorders in specialist

care in Sweden, we expect our model to be mainly applicable to a specialist care setting.

### Research aim
To predict the risk of suicidal behaviour among young individuals within 30 days of SSRI initiation, with a secondary analysis considering events within 90 days of initiation. We will use a two-stage approach in order to: (A) explore the performance of an ensemble prediction model using machine learning methods in Swedish register data, and to (B) develop a clinically applicable model with a restricted set of predictors where transparency and scalability is favoured.

## METHODS AND ANALYSIS
We will use a two-stage approach to produce a prediction model:

A. *Broad predictor model*: a relatively broad set of predictors is included in a number of different machine learning base models, the predictions of which are then combined to produce an ensemble prediction.
B. *Restricted predictor model:* a restricted set of top predictors identified from model A is included in a number of different machine learning models, which are compared in terms of predictive performance and transparency.

### Data source
We will obtain data from Swedish national registers, where linkage of records is based on unique personal identification numbers.[17] The Prescribed Drug Register—which includes all dispensed medications since July 2005—will be used to extract information on psychotropic medication receipt, including the initiation of SSRI medication.[18] The Total Population Register, which includes demographic information on Swedish residents since 1968, will be used to extract sex and country of birth for an individual.[19] The National Patient Register, documenting discharge diagnoses from inpatient and outpatient care since 1973 and 2001, respectively, will be used for information on psychiatric diagnoses, including suicide attempts.[20] The Cause of Death Register will be used to extract information on death by suicide.[21] The Longitudinal Integrated Database for Health Insurance and Labour Market Studies register—which includes information on socioeconomic variables for individuals aged 16 or above since 1990 and for those aged 15 or above since 2010—will be used to extract information on the highest commenced education between individuals and their parents.[22] The multigeneration register, including information on family relations,[23] will be used to link individuals to their parents, whose history of psychiatric disease and of death by suicide can then be extracted from the National Patient Register and Cause of Death register, respectively.

### Inclusion criteria
We will include all individuals aged 8–24 years who were born in Sweden and who initiated an SSRI after a wash-out

period of 365 days free from any antidepressants during 1 January 2007 to 31 December 2020. The first initiation event per individual will be included.

### Patient and public involvement

We have not involved patients in planning or executing this study. The research is based on observational register data, meaning no participant recruitment was necessary.

### Outcome ascertainment

Based on previous work, we will consider any International Classification of Diseases 10th revision (ICD-10) diagnosis code X60-X84 (suicidal behaviour of known intent) and Y10-Y34 (suicidal behaviour of unknown intent).[24 25] Only diagnoses made in specialist care are considered as our data linkage does not include primary care diagnoses.

We will consider suicide attempts or death by suicides (suicidal behaviour) within 30 days of SSRI initiation. As secondary analyses, we will (a) consider only suicidal behaviour of known intent within 30 days, and (b) consider suicidal behaviour within 90 days of SSRI initiation.

### Predictors
#### Broad predictor model

We will include a broad range of predictors based on the availability in our register linkage.

The predictors include: sex; age at SSRI initiation; history of mental and behavioural disorder diagnoses, previous suicidal behaviour and overdoses; psychotropic medications; highest commenced education between the index person and their parents in the year before SSRI initiation; prescriber practice type; parental history of mental and behavioural disorder diagnoses, suicidal behaviour and overdoses (same codes as in individuals); and family history of death by suicide. See tables 1 and 2 for diagnosis definitions, and table 3 for medication definitions.

All predictors aside from age will be considered as categorical variables. If there is evidence for it, we will consider accounting for non-linear effects of age in the elastic net logistic regression (EN) model.

### Restricted predictor model

We will include the following three predictors to ensure face-value validity of the prediction model: age, sex and history of previous suicide attempt. We will identify the top 30 predictors that contributed most to predictive performance in each of the base models in stage A (see section Model development). We will pool these predictors and include those deemed important based on clinical input, alongside the three face-value predictors.

### Missing values

We will exclude predictors where more than 20% of values are missing. In remaining predictors, if it is deemed that there is missingness at random, we will consider multiple imputation methods.[26]

| Table 1 | Predictors included in the broad predictor model, by register |
|---|---|
| **Total population register** | |
| Sex | |
| Native born | |
| **National patient register** | |
| Psychiatric diagnosis within 3 months before initiation | See table 2 for diagnosis subdivisions. |
| Psychiatric diagnosis >3 months before initiation | |
| **Prescribed drug register** | |
| Age at initiation | 1 year bands. |
| Prescriber practice | Primary/specialist/psychiatric specialist. |
| Psychiatric drug receipt within 3 months before initiation | See table 3 for medication subdivisions. |
| Psychiatric drug receipt >3 months before initiation | |
| **LISA** | |
| Highest commenced education of index individual's parents | Primary school, secondary school, university/vocational school. |
| **Multigeneration register+national patient register** | |
| Family history of psychiatric disease among the index individual's parents: lifetime before initiation | See table 2 for diagnosis subdivisions. |
| **Multigeneration register+cause of death register** | |
| Family history of death by suicide among index individual's parents: lifetime before initiation | All ICD-10 X60-X84; Y10-Y34. |
| ICD-10, International Classification of Diseases, 10th revision; LISA, Longitudinal Integrated Database for Health Insurance and Labour Market Studies. | |

**Table 2** Diagnoses codes included for predictors

| Diagnosis | ICD-10 code categorisation |
|---|---|
| Mental and behavioural disorders | |
| Organic mental disorder | F00-F09 |
| Substance use disorder | F10-19 |
| Schizophrenia and psychotic disorder | F20-F29 |
| Manic episode | F30 |
| Bipolar disorder | F31 |
| Depression | F32, F33 |
| Persistent mood disorder | F34 |
| Other or unspecified mood disorders | F38, F39 |
| Phobic anxiety disorder | F40.0-F40.2 |
| Other anxiety disorders | F41.0-F41.1 |
| OCD | F42 |
| Reaction to severe stress, and adjustment disorders | F43 |
| Anorexia nervosa | F500, F501 |
| Bulimia nervosa | F502, F503 |
| Non-organic sleep disorders | F51 |
| Dissocial personality disorder or emotionally unstable personality disorder | F602, F603 |
| Gender identity disorders | F64 |
| Mental retardation | F70-F79 |
| Autism | F840, F841, F845, F848, F849 |
| Hyperkinetic disorder | F90 |
| Conduct disorders | F91 |
| Suicidal behaviour | X60-X84, Y10-Y34 |
| Overdose | T36-51, X40-49 |

ICD-10, International Classification of Diseases, 10th revision; OCD, Obsessive-Compulsive Disorder.

**Table 3** ATC codes included for predictors

| Medication | ATC code |
|---|---|
| Opioids and pain medication | N02A |
| Antiepileptic drugs | N03A |
| Antipsychotics | N05A |
| Anxiolytics, hypnotics and sedatives | N05B excluding N05BA, N05C |
| Benzodiazepines | N05BA |
| Antidepressants, excluding SSRIs | N06A excluding N06AB |
| ADHD medication | N06B |
| Drugs used in addiction disorders | N07B |

ADHD, attention deficit hyperactivity disorder; ATC, Anatomical Therapeutic Chemical; SSRI, selective serotonin reuptake inhibitor.

## Model validation and goodness of fit

Overall classification performance using area under the receiver operating characteristic curves (AUC) will be estimated for all models; 95% CIs will be calculated using DeLong's method.[12 27]

For the broad variable search model and for the selected restricted predictor model, we will assess the predictive performance of the trained models using: Concordance Index, Brier Score and a calibration plot.[12 14]

For the broad variable search model and for the selected restricted predictor model, we will also asses sensitivity, specificity, positive and negative predictive values at the risk threshold of 1%, given that this is the approximate prevalence of suicidal behaviour within 30 days after SSRI initiation in Swedish registers.[25] That is, any child,

adolescent or young adult with a predicted risk greater than 1% have a risk that is higher than what is expected in the population, and can therefore be considered 'high-risk'.[14 28] We will also report calibration performance at a 2% and 5% risk cut-off.

The impact of different risk score cut-offs could be further evaluated in future work—however, continuous risk scores could provide a more flexible idea of individual risk as compared with risk cut-offs.[28]

We will follow the Transparent Reporting of a multivariable prediction model for Individual Prognosis or Diagnosis (TRIPOD) guidelines.[29]

## Model development

Throughout the paper, we will employ different machine learning models: XGBoost (XG), neural net (NN), EN and support vector machines (SVM).[12 30]

We will develop all models in a sample ('training sample') using all data from 1 January 2007 to 31 December 2017. We will apply shrinkage procedures where appropriate, for example, by employing an EN.

We will validate all models in a sample ('testing sample') using all data from 1 January 2018 to 31 December 2020. That is, no random sampling will be performed for either training or testing sample.

As a secondary validation, we will validate the final restricted model in the period 1 January 2018 to 28 February 2020 to see if the model performance is influenced by the COVID-19 pandemic.

### Broad predictor model

For each model XG, NN, EN and SVM, we will identify hyperparameters by a grid search with 10-fold stratified cross-validation, ensuring an approximately equal proportion of individuals with the outcome in each fold. We will use the AUC as the evaluation criterion for hyperparameter tuning[30]—that is, we will choose the set of hyperparameters that maximise the AUC for each model.

After model tuning, we will combine the predictions from all the different models. The class probabilities of each model will be summed and averaged—the final prediction will consist of the class with the highest average probability (ie, we will produce an ensemble prediction). In effect, the predictions of each model will be weighted equally.

Finally, the 30 predictors that are most important (ie, contributed most to the predictive performance) in each of the base models (when applied in the training data) will be extracted using permutation-based feature importance, if computationally feasible. The pooled list of these predictors will then be considered for inclusion in the restricted predictor model based on clinical input and the available evidence base on the risk factors; the original set of pooled predictors will be presented, and we will describe our reasoning for the final predictor inclusion in the paper.

If permutation-based feature importance is not computationally feasible, we will consider the most important predictors based on the predictor importance score from the XG model and absolute beta coefficients from the EN model.

### Restricted predictor model

For each model XG, NN, EN and SVM, we will identify hyperparameters by a grid search with 10-fold stratified cross-validation, ensuring an approximately equal proportion of individuals with the outcome in each fold. We will use the AUC as the evaluation criterion for hyperparameter tuning.[30]

We will then compare the predictive performance of the different models when including the restricted set of predictors in the testing sample, using the DeLong method to assess statistical significance of the difference. We will also produce an ensemble prediction in the same way as described for the broad predictor model. If there is no model that performs better than the others in terms of AUC score (including the ensemble model) and calibration performance, we will consider the EN model as the final restricted predictor model.[15] If there is a model with statistically significantly better AUC performance compared with the others (including the ensemble model), this model will be selected as the final restricted predictor model.

We favour the EN model in absence of clear superiority because this multivariate regression approach allows more transparency: the intercept and each predictor coefficient can be reported, allowing a prediction score for each study participant to be easily calculated by clinicians. This intercept of the model, which is a transformation of the baseline risk in the population, can also be changed to fit the particular population considered. In other words, an EN model would allow for greater transparency and scalability, including easy dissemination of the model as an online tool.[14]

Given the parameter selection inherent in an EN model, it is possible that even in the broad predictor model step it produces a model with fewer than 30 predictors. If this model already at the stage of the broad predictor model has a predictive performance superior to or equal to the other models, we will consider bypassing stage B of this protocol and using this model as the final model.

In the final model, we will assess the prediction performance in terms of the AUC in the subset of individuals who received a diagnosis of depression (ICD-10 code F32 or F33) in the previous 90 days. In the group of individuals aged 18–24 years, we will also consider testing the model performance separately for individuals initiating their SSRI in primary and specialist care settings.

### Generalisability

Predictors have been defined in such a way that it will be possible to apply the models in other countries and find comparable predictors, for example, by using standard ICD-10 and Anatomical Therapeutic Chemical (ATC) codes to define diagnosis and medication predictors, respectively. While our time-based validation allows some generalisability over time in the Swedish context, external validation from other national settings could be a next step.

## ETHICS AND DISSEMINATION

The research has been approved by the Swedish Ethical Review Authority (decision number 2020–06540). The use of anonymised register data waives the need to secure informed consent.[31] We will disseminate our findings by publishing in peer-reviewed open-access journals, and presenting at international conferences.

**Contributors** TL, ZC and SF conceptualised the study. TL drafted the protocol. All authors provided critical input on protocol content and the protocol write-up. All authors fulfil authorship criteria as set out by International Committee of Medical Journal Editors (ICMJE).

**Funding** This work was supported by the Swedish Research Council (grant number 2018-02213), and by the National Institute for Health and Care Research (NIHR) Applied Research Collaboration Oxford and Thames Valley at Oxford Health NHS Foundation Trust (grant number N/A). The views expressed are those of the authors and not necessarily those of the National Health Service, the NIHR or the Department of Health and Social Care. SV is funded by Forte (Swedish Research Council for Health, Working Life and Welfare, Postdoctoral Fellowship, grant number 2022-00824).

**Competing interests** None declared.

**Patient and public involvement** Patients and/or the public were not involved in the design, or conduct, or reporting, or dissemination plans of this research.

**Patient consent for publication** Not applicable.

**Provenance and peer review** Not commissioned; externally peer reviewed.

**ORCID iD**
Tyra Lagerberg http://orcid.org/0000-0003-2480-0216

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
