## [Reviewer comments · BMJ Open]

ARTICLE DETAILS

TITLE (PROVISIONAL)	Predicting risk of suicidal behaviour after initiation of selective serotonin reuptake inhibitors in children, adolescents, and young adults: protocol for development and validation of clinical prediction models
AUTHORS	Lagerberg, Tyra; Virtanen, Suvi; Kuja-Halkola, Ralf; Hellner, Clara; Lichtenstein, Paul; Fazel, Seena; Chang, Zheng

VERSION 1 – REVIEW

REVIEWER	Stuke, Heiner Charité Universitätsmedizin Berlin
REVIEW RETURNED	21-Apr-2023

GENERAL COMMENTS	The authors describe an interesting and clinically relevant study on predicting the risk of suicidal behavior after initiation of an SSRI in individuals younger than 25 with machine learning and data from different Swedish registers. Overall, the methods described seem sound to me. My main issue is that I think that the design as currently planned does not allow for a distinction between general prognostic factors and specific predictors of increased suicide risk after SSRI initiation, which would be crucial for treatment decisions. In addition, I have some minor proposals regarding the planned methods. Major point: - The authors aim to develop a predictive tool for the risk of suicidal behavior after starting an SSRI in young people. Such a tool would certainly be of clinical relevance, as it could allow risk stratification after SSRI initiation and thus, for example, in the outpatient sector, might allow the frequency of follow-up visits to be adapted to the individual risk. However, as it is currently planned, I do not believe that the tool can influence the decision in which young patients an SSRI should (not) be started. This is because the effect of the SSRI in the risk predictions cannot be determined when no control group without SSRI initiation is studied. For example, if the tool predicts a very high risk of suicidal behavior for a patient, it is not clear whether this risk is in any way lower when not starting an SSRI. In other words, general prognostic factors (independent of treatment) and specific predictive factors (dependent on treatment) cannot be distinguished. My concern is that clinicians who read the paper superficially may not grasp this difference and think that high predicted risk is necessarily an argument against SSRI initiation. For instance, the sentence from the introduction “[...] producing a model to assess which individuals are more or less likely to engage in suicidal behaviour after initiating SSRI medication could be an important
---

	aid in clinical decision-making [...]” suggests that the tool could be used to make decisions about the initiation of an SSRI. If the authors agree that the tool, as currently planned, does not allow to determine in which patients there is an additional risk of suicidal behavior after SSRI initiation, there are two options, as I see it. Either (and preferably, if possible), a second data set with as well-matched patients as possible without SSRI initiation is extracted. Then, the additional risk from SSRI initiation could be estimated by comparing risk predictions between the SSRI and no-SSRI models (similar to the personalized advantage index). If that is not possible, I would at least make a stronger point in the text that the tool predicts suicide risk after SSRI initiation but not suicide risk from SSRI initiation. Minor points:  - Probably, the outcome classes will be highly imbalanced (few patients with suicidal behavior). Does this have to be taken into account in the analysis (i.e., by using stratified cross validation)? - Authors must ensure that the determination of feature importance in the broad model, which determines the predictor selection of the constrained model, is made only in the training data. Otherwise, it would be an improper model development and validation on the same data. I assume this is the plan anyway, but the manuscript is not entirely clear on this. - Some information on the selection of predictors from the broad model is missing. It is mentioned that in the linear model absolute coefficient values and for boosting machines predictor importance scores (I assume this refers to impurity based feature importance) should be used. What about for neural networks and support vector machines? The authors might consider whether permutation based feature importances are a general option. There is also mention of clinical input for predictor selection. Again, more details would be helpful.
--	---

REVIEWER	Bhavsar, Vishal Institute of Psychiatry Department of Health Service and Population Research, Department of Health Services and Population Research
REVIEW RETURNED	11-May-2023

GENERAL COMMENTS	Thank you. Restriction of the validation data by depression diagnosis could be referred to in the abstract and introduction I think, given SSRI are used for a range of indications. At "All models will be tested..." please can you clarify whether this is internal validation? So as to align it with CPM literature/TRIPOD- this would then allow you to state that external val. would be the next step(if it is). An intuitive explanation of ensemble models would be helpful, somewhere. It's a good introduction and an important problem but could you say a bit more about the different implementation contexts and how much you know about the different data needs/clinical scenarios- if there is anything in the literature/principles from clinical experience. "We will include all individuals aged 8-24 years who were born in Sweden and who initiated an SSRI after a wash-out period of 365 days free from any antidepressants during 1st January 2008 to 31st December 2020. The first initiation event per individual will be included."
---

	Re. the above, it would be helpful to discuss how this limit could change the clinical applicability of a model- e.g. you would be expecting clinicians to filter this group into the model through a prior question(have you been free of SSRI for one year prior to this initiation). I was slightly unclear about the meaning of: "For the final implementation of the model in clinical practice, we will provide a continuous risk score rather than a risk cut-off to provide a more flexible and individually-tailored aid to clinical decision-making." - it doesn't sound like you would be moving directly to clinical practice, so the first clause sounds a bit out of place- it might be better to say that you could evaluate the different impact of cut-offs vs. risk scores in later stages of the work(so that you are not committed to cutoffs). Of course a number of studies evaluate the comparative performance of different models, but what is the evidence(previous studies) that this kind of comparison and model selection is useful? You write the benefits in principle a priori of LR, so what would e.g. an NN model do to improve this? The second and third sentences under Model Development are really helpful and the terminology could be included in the abstract to clarify the stages of the study. The sentence under Generalizability is reasonable but could you make this more concrete- how will you pick definitions which are generalizable, or are there decisions you have already made which reflect this purpose? You don't describe existing risk factor or other clinical epidemiological evidence on suicidal behaviour in SSRI- exposed young people. Could you do this, if this is available? Presumably this should inform predictor selection- i.e. what do we already know about what predicts this? Tables 2 and 3 indicate a large number of parameters. Will modelling account for/address overfitting in the testing stage, e.g. through shrinkage procedures?
--	--

VERSION 1 – AUTHOR RESPONSE

Reviewer: 1

Dr. Heiner Stuke, Charité Universitätsmedizin Berlin

Comments to the Author:

The authors describe an interesting and clinically relevant study on predicting the risk of suicidal behavior after initiation of an SSRI in individuals younger than 25 with machine learning and data from different Swedish registers. Overall, the methods described seem sound to me. My main issue is that I think that the design as currently planned does not allow for a distinction between general prognostic factors and specific predictors of increased suicide risk after SSRI initiation, which would be crucial for treatment decisions. In addition, I have some minor proposals regarding the planned methods.

1.1) Major point:

- The authors aim to develop a predictive tool for the risk of suicidal behavior after starting an SSRI in young people. Such a tool would certainly be of clinical relevance, as it could allow risk stratification after SSRI initiation and thus, for example, in the outpatient sector, might allow the frequency of follow-up visits to be adapted to the individual risk. However, as it is currently planned, I do not believe that the tool can influence the decision in which young patients an SSRI should (not) be started. This is because the effect of the SSRI in the risk predictions cannot be determined when no control group without SSRI initiation is studied. For example, if the tool predicts a very high risk of suicidal behavior for a patient, it is not clear whether this risk is in any way lower when not starting an SSRI. In other words, general prognostic factors (independent of treatment) and specific predictive factors (dependent on treatment) cannot be distinguished.

My concern is that clinicians who read the paper superficially may not grasp this difference and think that high predicted risk is necessarily an argument against SSRI initiation. For instance, the sentence from the introduction “[...] producing a model to assess which individuals are more or less likely to engage in suicidal behaviour after initiating SSRI medication could be an important aid in clinical decision-making [...]” suggests that the tool could be used to make decisions about the initiation of an SSRI.

If the authors agree that the tool, as currently planned, does not allow to determine in which patients there is an additional risk of suicidal behavior after SSRI initiation, there are two options, as I see it. Either (and preferably, if possible), a second data set with as well-matched patients as possible without SSRI initiation is extracted. Then, the additional risk from SSRI initiation could be estimated by comparing risk predictions between the SSRI and no-SSRI models (similar to the personalized advantage index). If that is not possible, I would at least make a stronger point in the text that the tool predicts suicide risk after SSRI initiation but not suicide risk from SSRI initiation.

We thank the reviewer for this important point. Our prediction model is not aiming to provide answers regarding causality – that is, it cannot answer questions about to what extent SSRIs influence the risk of suicidal behavior, which might be used to direct clinicians regarding whether to prescribe a patient the medication or not. Rather, the aim of the model is to identify individuals who have particularly elevated risk for suicidal behavior during SSRI treatment, regardless of causality. This is clinically important given that many individuals receiving SSRIs are already a high-risk population, meaning clinicians may need input on which patients to pay more attention to.

We have clarified that our model is not developed for causal prediction in the text:

Intro:

“Regardless of causality, producing a model to assess which individuals, out of those initiating an SSRI, are more or less likely to engage in suicidal behaviour could be an important aid in clinical decision-making in terms of determining which individuals will require more clinical oversight.”

(...)

“It should be noted that our aim is not to determine which individuals should or should not be treated with an SSRI, but to identify whom, among those who do initiate an SSRI, is at particularly elevated risk of suicidal behaviour.”

To elaborate somewhat on the interesting first option suggested by the reviewer: the reviewer here refers to causal prediction modelling. Such prediction models are gaining traction in the literature.¹ However, it is a relatively new area.

The reviewer mentions the “personalized advantage index”,² where the results from two randomized controlled trials (RCTs) were used to produce a causal model for the impact of antidepressant medication versus psychotherapy. In an observational setting, this could be approximated by something akin to what the reviewer suggests – setting up a target trial framework to compare those who do and do not initiate SSRIs (e.g. Prosperi et al. 2020³ – in fact, this article argues against the usefulness of data-driven approaches). However, a causal prediction model ideally would require access to several further clinically important features than we have access to – for example, depression severity – in order to make sure that those who are and are not prescribed an SSRI are truly exchangeable and counterfactual predictions can be calculated. In our setting, we therefore aim

to look only at those who have been indicated for SSRI treatment in order to produce risk stratification in this group, even if the model does not give us information on the effect of SSRIs in themselves on suicidal behavior risk.

Minor points:

1.2) - Probably, the outcome classes will be highly imbalanced (few patients with suicidal behavior).

Does this have to be taken into account in the analysis (i.e., by using stratified cross validation)?

We thank the reviewer for this important consideration. This is when we ensure there is approximately the same outcome prevalence in each fold of the cross-validation. Stratified cross-validation has been shown to reduce bias and variance in accuracy estimation and model selection.⁴ We have included this feature in the prediction protocol:

“For each model XG, NN, EN, and SVM, we will identify hyperparameters by a grid search with 10-fold stratified cross-validation, ensuring an approximately equal proportion of individuals with the outcome in each fold.”

1.3) - Authors must ensure that the determination of feature importance in the broad model, which determines the predictor selection of the constrained model, is made only in the training data.

Otherwise, it would be an improper model development and validation on the same data. I assume this is the plan anyway, but the manuscript is not entirely clear on this.

We appreciate the request to clarify this in the manuscript – we have amended to emphasize that this is what we intend to do:

“Finally, the 30 predictors that are most important (i.e. contributed most to the predictive performance) in each of the base models (when applied in the training data) will be extracted using permutation based feature importance if computationally feasible.”

1.4) - Some information on the selection of predictors from the broad model is missing. It is mentioned that in the linear model absolute coefficient values and for boosting machines predictor importance scores (I assume this refers to impurity based feature importance) should be used. What about for neural networks and support vector machines? The authors might consider whether permutation based feature importances are a general option. There is also mention of clinical input for predictor selection. Again, more details would be helpful.

We thank the reviewer for this suggestion. We will instead use a permutation based feature importance for the models, if computationally feasible. As part of this, we will also consider variables that are highly correlated, and possibly cluster these to avoid underestimating the importance of correlated features.⁵

“Finally, the 30 predictors that are most important (i.e. contributed most to the predictive performance) in each of the base models (when applied in the training data) will be extracted using permutation based feature importance if computationally feasible.”

“If permutation-based feature importance is not computationally feasible, we will consider the most important predictors based on the predictor importance score from the XG model, and absolute beta coefficients from the EN model.”

As regards clinical input, we will ask clinicians to assess the list of the pooled predictors produced in this step. We will present the pooled set of predictors and describe our reasoning for which to include in the final paper to make this decision-making process transparent. We commit to such transparency with the following amendment:

“The pooled list of these predictors will then be considered for inclusion in the restricted predictor model based on clinical input and the available evidence base on the risk factors; the original set of pooled predictors will be presented, and we will describe our reasoning for the final predictor inclusion in the paper.”

Reviewer: 2

Dr. Vishal Bhavsar, Institute of Psychiatry Department of Health Service and Population Research

Comments to the Author:

Thank you. Restriction of the validation data by depression diagnosis could be referred to in the abstract and introduction I think, given SSRI are used for a range of indications.

We thank the reviewer for this point. We have included a reference to this in the abstract:

“We will assess the final model performance in individuals receiving a depression diagnosis within 90 days before SSRI initiation.”

2.1) At "All models will be tested..." please can you clarify whether this is internal validation? So as to align it with CPM literature/TRIPOD- this would then allow you to state that external val. would be the next step(if it is).

Many thanks for this comment. Internal validation is when validation of the model is performed on new individuals that come from the same population as the one the model was developed in. We use time-based validation in this study, which can be considered to be “in between” internal and external validation.⁶ We have commented on this in the paper:

“While our time-based validation allows some generalizability over time in the Swedish context, external validation from other national settings could be a next step.”

2.2) An intuitive explanation of ensemble models would be helpful, somewhere.

Many thanks for this suggestion – we have included a sentence to address this:

“After model tuning, we will combine the predictions from all the different models. The class probabilities of each model will be averaged – the final prediction will consist of the class with the highest average probability (that is, we will produce an ensemble prediction). In effect, the predictions of each model will be weighted equally.”

2.3) It's a good introduction and an important problem but could you say a bit more about the different implementation contexts and how much you know about the different data needs/clinical scenarios- if there is anything in the literature/principles from clinical experience.

At this stage, the model is more explorative and more work would have to be conducted before clinical implementation. However, we would hope that it could be used alongside other tools for risk assessment to help stratify individuals based on suicidal behavior risk. For example, it could be useful for a clinician when they are prescribing an SSRI to determine the follow-up schedule of a patient. An ideal scenario would be where the risk score is automatically generated for a patient based on their past contacts with the healthcare service, in which case the broad predictor model could be implemented in clinical practice, if its performance is superior to that of the restricted predictor model. The more likely scenario is that the clinician would have to interview the patient to obtain information on predictors. In that case, the restricted predictor model would be more feasible to implement. Regarding context of model implementation, in principle it is a “generalist” model in the sense that both individuals who are prescribed their SSRIs in specialist or primary care setting are included. We have specialist/outpatient setting as a predictor in our model. However, in practice individuals under age 18 are routinely treated in child and adolescent psychiatry (specialist) care settings for mood disorders – the main indication for SSRI treatment. We also only have data on diagnostic information from a specialist setting. Our model may therefore be most applicable to individuals accessing specialist healthcare, although we may additionally test the model performance separately for individuals receiving their SSRIs from a primary and specialist care setting among those aged 18 and above – in a prior study,⁷ around 40% of individuals aged 18 or above received their antidepressant from primary care.

We have addressed the setting of the model in the introduction:

“Given that children under age 18 are routinely treated for psychiatric disorders in specialist care in Sweden, we expect our model to be mainly applicable to a specialist care setting.”

We have also included a suggested test of model performance separately in primary and specialist care settings in individuals aged 18 or above:

“In the group of individuals aged 18-24 years, we will also consider testing the model performance separately for individuals initiating their SSRI in primary and specialist care settings.”

With regards to clinician preferences, previous studies indicate that they prefer to understand why a given patient is flagged as either high or low-risk, especially when the risk classification seems counterintuitive, which might suggest that more transparent models (such as logistic regression) are favourable.⁸

We have elaborated slightly on this in the introduction:

“Some previous prediction studies have incorporated a wide variety of available predictors in order to harness the potential of large routinely collected electronic health records.⁹⁻¹¹ For example, a previous study using Swedish register data was able to predict suicide following a psychiatric inpatient visit with a relatively high level of discrimination performance.¹⁰ However, this approach results in large models that may not be practical to implement in real-life settings where clinicians do not have access to routinely collected data. In such a setting, a clinician might want a restricted number of predictors to input into a risk prediction model that is simple, scalable, and transparently operationalized, for example as an online tool.^{12 13} Previous research also suggests that clinicians want to understand why specific individuals are assigned a specific risk score, in particular when the risk classification is counterintuitive,⁸ which may favour the latter type of model.”

2.4) "We will include all individuals aged 18-24 years who were born in Sweden and who initiated an SSRI after a wash-out period of 365 days free from any antidepressants during 1st January 2008 to 31st December 2020. The first initiation event per individual will be included."

Re. the above, it would be helpful to discuss how this limit could change the clinical applicability of a model- e.g. you would be expecting clinicians to filter this group into the model through a prior question (have you been free of SSRI for one year prior to this initiation).

We appreciate this point. We will aim to include such a discussion in the final paper.

2.5) I was slightly unclear about the meaning of: "For the final implementation of the model in clinical practice, we will provide a continuous risk score rather than a risk cut-off to provide a more flexible and individually-tailored aid to clinical decision-making." - it doesn't sound like you would be moving directly to clinical practice, so the first clause sounds a bit out of place- it might be better to say that you could evaluate the different impact of cut-offs vs. risk scores in later stages of the work(so that you are not committed to cutoffs).

Many thanks for this useful point. We have amended as suggested:

“The impact of different risk score cut-offs could be further evaluated in future work – however, continuous risk scores could provide a more flexible idea of individual risk as compared to risk cut-offs.¹⁴”

2.6) Of course a number of studies evaluate the comparative performance of different models, but what is the evidence(previous studies) that this kind of comparison and model selection is useful? You write the benefits in principle a priori of LR, so what would e.g. an NN model do to improve this? We appreciate this point. Several previous studies highlight the fact that, for this type of prediction and in this type of data, logistic regression models tend to perform similarly compared to more complex models such as neural networks.¹⁵

However, there are previous examples in a Swedish register setting of an ensemble of different machine-learning models producing more accurate prediction of suicidal behavior following psychiatric discharge than individual models.¹⁰ We therefore seek to evaluate whether we can achieve a higher prediction accuracy by incorporating several models.

2.7) The second and third sentences under Model Development are really helpful and the terminology could be included in the abstract to clarify the stages of the study.

We appreciate this point.

The second and third sentences under Model Development relate to the training and testing sample of the models.

We have attempted to clarify the training and testing sets in the “Methods and analysis” section of the abstract:

“The training and testing samples will consist of data from 2007-2017 and 2018-2020, respectively.”

2.8) The sentence under Generalizability is reasonable but could you make this more concrete- how will you pick definitions which are generalizable, or are there decisions you have already made which reflect this purpose?

Thanks for this suggestion. We have already defined our predictors in a generalizable way, by using internationally recognized ICD-10 codes and ATC codes for diagnoses and medications, respectively, and by using commonly used classifications for clinical and sociodemographic characteristics (e.g. defining educational attainment by primary school/high school/university).

We have added a sentence to reflect this:

“Predictors have been defined in such a way that it will be possible to apply the models in other countries and find comparable predictors, for example by using standard ICD-10 and ATC codes to define diagnosis and medication predictors, respectively.”

2.9) You don't describe existing risk factor or other clinical epidemiological evidence on suicidal behaviour in SSRI- exposed young people. Could you do this, if this is available? Presumably this should inform predictor selection- i.e. what do we already know about what predicts this?

We appreciate this important point. Part of the purpose of the paper is to identify important predictors for suicidal behavior in the broad predictor model step. However, some will be included in all models to ensure face value validity – e.g. past suicidal behavior, which has been flagged as a key predictor of further suicidal behavior, alongside sex and age. Another class of key predictor is psychiatric diagnoses.¹⁶ Further important predictors include sexual orientation,¹⁷ exposure to racism,¹⁸ suicides among relatives or friends,¹⁹ illicit drug use,¹⁹ academic pressure,¹⁹ bullying and social isolation,¹⁹ sleep disturbances,²⁰ and more. Some of these risk factors we have access to in our specific register linkage, others we do not.

We will discuss the evidence base for important predictors in the final study in order to give context to the clinical input on what predictors to include in the restricted predictor model. We commit to transparency regarding the predictor selection in the restricted predictor model with the following statement:

“The pooled list of these predictors will then be considered for inclusion in the restricted predictor model based on clinical input and the available evidence base on the risk factors; the original set of pooled predictors will be presented, and we will describe our reasoning for the final predictor inclusion in the paper.”

2.10) Tables 2 and 3 indicate a large number of parameters. Will modelling account for/address overfitting in the testing stage, e.g. through shrinkage procedures?

This is an important point.

We aim to apply shrinkage procedures where appropriate – for example, we will apply elastic net logistic regression rather than logistic regression. We have exchanged any mention of LR with elastic net logistic regression (EN). We have noted this in the protocol:

“We will apply shrinkage procedures where appropriate, for example by employing an elastic net logistic regression.”

Given the parameter selection inherent in an EN model, it is possible that even in the broad predictor model step it produces a model with 30 or fewer predictors. If this model already at the stage of the broad predictor model has a predictive performance superior to or equal to the other models, we will

consider bypassing stage B of this protocol and using this model as the final model.

We have included the above paragraph in the model development section:

“Given the parameter selection inherent in an EN model, it is possible that even in the broad predictor model step it produces a model with fewer than 30 predictors. If this model already at the stage of the broad predictor model has a predictive performance superior to or equal to the other models, we will consider bypassing stage B of this protocol and using this model as the final model.”

To further reduce the probability of overfitting, we have reduced the number of predictors included (see Tables 1 and 2). We have also decided to include information on SSRI initiators from a wider range of years (January 2007 to December 2020) in order to ensure a greater number of outcome events – we have changed all mentions of the study period to reflect this.

Reviewer: 1

Competing interests of Reviewer: None

Reviewer: 2

Competing interests of Reviewer: None to declare.

References

1. Lin L, Sperrin M, Jenkins DA, et al. A scoping review of causal methods enabling predictions under hypothetical interventions. *Diagnostic and Prognostic Research* 2021;5(1):3. doi: 10.1186/s41512-021-00092-9
2. DeRubeis RJ, Cohen ZD, Forand NR, et al. The Personalized Advantage Index: translating research on prediction into individualized treatment recommendations. A demonstration. *PloS one* 2014;9(1):e83875.
3. Proserpi M, Guo Y, Sperrin M, et al. Causal inference and counterfactual prediction in machine learning for actionable healthcare. *Nature Machine Intelligence* 2020;2(7):369-75. doi: 10.1038/s42256-020-0197-y
4. A study of cross-validation and bootstrap for accuracy estimation and model selection. *Ijcai*; 1995. Montreal, Canada.
5. SciKitLearn. Permutation feature importance 2023 [Available from: https://scikit-learn.org/stable/modules/permutation_importance.html accessed June 4th 2023.
6. Ramspek CL, Jager KJ, Dekker FW, et al. External validation of prognostic models: what, why, how, when and where? *Clinical Kidney Journal* 2020;14(1):49-58. doi: 10.1093/ckj/sfaa188
7. Lagerberg T, Molero Y, D'Onofrio BM, et al. Antidepressant prescription patterns and CNS polypharmacy with antidepressants among children, adolescents, and young adults: a population-based study in Sweden. *Eur Child Adolesc Psychiatry* 2019;28(8):1137-45. doi: 10.1007/s00787-018-01269-2 [published Online First: 2019/01/20]
8. Yarborough BJH, Stumbo SP, Schneider J, et al. Clinical implementation of suicide risk prediction models in healthcare: a qualitative study. *BMC psychiatry* 2022;22(1):789.
9. Gradus JL, Rosellini AJ, Horváth-Puhó E, et al. Prediction of sex-specific suicide risk using machine learning and single-payer health care registry data from Denmark. *JAMA psychiatry* 2020;77(1):25-34.
10. Chen Q, Zhang-James Y, Barnett EJ, et al. Predicting suicide attempt or suicide death following a visit to psychiatric specialty care: A machine learning study using Swedish national registry data. *PLoS medicine* 2020;17(11):e1003416.
11. Gradus JL, Rosellini AJ, Horváth-Puhó E, et al. Predicting sex-specific nonfatal suicide attempt risk using machine learning and data from Danish national registries. *American journal of epidemiology* 2021;190(12):2517-27.
12. Fazel S, Wolf A, Larsson H, et al. The prediction of suicide in severe mental illness: development and validation of a clinical prediction rule (OxMIS). *Translational psychiatry* 2019;9(1):1-10.
13. Whiting D, Fazel S. How accurate are suicide risk prediction models? Asking the right questions for clinical practice. *Evidence-based mental health* 2019;22(3):125-28.

14. Wynants L, van Smeden M, McLernon DJ, et al. Three myths about risk thresholds for prediction models. BMC medicine 2019;17(1):1-7.
15. Tate AE, McCabe RC, Larsson H, et al. Predicting mental health problems in adolescence using machine learning techniques. PloS one 2020;15(4):e0230389.
16. Fazel S, Runeson B. Suicide. New England Journal of Medicine 2020;382(3):266-74. doi: 10.1056/NEJMra1902944
17. Amitai M, Apter A. Social aspects of suicidal behavior and prevention in early life: a review. International journal of environmental research and public health 2012;9(3):985-94.
18. Rudes G, Fantuzzi C. The association between racism and suicidality among young minority groups: a systematic review. Journal of transcultural nursing 2022;33(2):228-38.
19. Rodway C, Tham S-G, Ibrahim S, et al. Suicide in children and young people in England: a consecutive case series. The Lancet Psychiatry 2016;3(8):751-59.
20. Liu J-W, Tu Y-K, Lai Y-F, et al. Associations between sleep disturbances and suicidal ideation, plans, and attempts in adolescents: a systematic review and meta-analysis. Sleep 2019;42(6)

VERSION 2 – REVIEW

REVIEWER	Stuke, Heiner Charité Universitätsmedizin Berlin
REVIEW RETURNED	04-Jul-2023

GENERAL COMMENTS	The authors adequately addressed all of my points. I recommend acceptance of the manuscript and wish you success in the study.
--

REVIEWER	Bhavsar, Vishal Institute of Psychiatry Department of Health Service and Population Research, Department of Health Services and Population Research
REVIEW RETURNED	13-Jul-2023

GENERAL COMMENTS	Thank you for your responses. I have nothing further to add.
--